# Evaluation of the QoS policy model of an ordinary 5G smart city cluster with predominant URLLC and eMBB traffic

**Viacheslav Kovtun**[1]*, **Krzysztof Grochla**[2], **Torki Altameem**[3], **Mohammed Al-Maitah**[3]

**1** Computer Control Systems Department, Vinnytsia National Technical University, Vinnytsia, Ukraine, **2** Internet of Things Group, Institute of Theoretical and Applied Informatics Polish Academy of Sciences, Gliwice, Poland, **3** Computer Science Department, Community College, King Saud University, Riyadh, Saudi Arabia

* kovtun_v_v@vntu.edu.ua

**Data Availability Statement:** All relevant data are within the paper and its Supporting Information files.

## Abstract

A typical element of the smart city's information and communication space is a 5G cluster, which is focused on serving both new and handover requests because it is an open system. In an ordinary 5G smart city cluster, Ultra-Reliable Low-Latency Communications (URLLC) and enhanced Mobile BroadBand (eMBB) traffic types prevail. The formation of an effective QoS policy for such an object (taking into account the potentially active slicing technology) is an urgent problem. As a baseline, this research considers a Quality of Service (QoS) policy with constraints for context-defined URLLC and eMBB classes of incoming requests. Evaluating the QoS policy instance defined within the framework of the basic concept requires the formalization of both a complete qualitative metric and a computationally efficient mathematical apparatus for its calculation. The article presents accurate and approximate methods of calculating such quality parameters as the probability of loss of typed requests and the utilization ratio of the communication resource, which depend on the implementation of the estimated QoS policy. At the same time, the original parametric space includes both fixed characteristics (amount of available communication resources, load according to request classes) and controlled characteristics due to the specifics of the implementation of the basic QoS concept. The paper empirically proves the adequacy of the presented mathematical apparatus for evaluating the QoS policy defined within the scope of the research. Also, in the proposed qualitative metric, a comparison of the author's concept with a parametrically close analogue (the well-known QoS policy scheme, which takes into account the phenomenon of reservation of communication resources), determined taking into account the reservation of communication resources, was made. The results of the comparison testify in favour of the superiority of the author's approach in the proposed metrics.

## 1 Introduction and state-of-the-art

The creation of the 5G platform was a response to the growing need to solve problems such as [1–3]: growth in mobile traffic, an increase in wireless network-connected devices number,

**Funding:** Torki Altameem (T. A.), Researchers Supporting Project no. RSP2023R503, King Saud University, Saudi Arabia.

**Competing interests:** The authors have declared that no competing interests exist.

and a need to reduce delays when introducing new services. The main means to solve this impressive list of problems are three leading technologies [4, 5]: Ultra-Reliable Low-Latency Communications (URLLC), enhanced Mobile BroadBand (eMBB), and massive Machine-Type Communications (mMTC). The mMTC technology is focused on serving the traffic of a huge number of applications and devices that do not demand latency. The URLLC technology is designed to support mission-critical communications with low latency, high security and reliability. Finally, eMBB technology aims to provide high data transfer rates. This issue is discussed in more detail in [2].

The most relevant platform for the real application of all the capabilities of the 5G platform is the information space of a smart city. To a first approximation, this space can be segmented into industrial, specialized and civil clusters that require wireless communication coverage. At the same time, it is civic clusters that dominate the information and communication landscape of the modern agglomeration. This circumstance allows us to further call the "civilian" 5G cluster ordinary. At the same time, typical information exchange in such a 5G cluster requires the implementation of URLLC (ensuring the information needs of critical infrastructure and institutions) and eMBB (ensuring the information needs of citizens and the service sector) technologies [6]. In this context, the urgent problem arises of providing high-quality service to the needs of users of an ordinary 5G cluster while simultaneously using URLLC and eMBB technologies.

Aspects of the joint use of URLLC and eMBB technologies are being studied by many teams of scientists. This is confirmed by current review articles [7–11] and the impressive list of specialized studies that are mentioned in them. The authors of articles [12, 13] propose to allocate resources for URLLC traffic, prioritizing the reliability of connections. In [12, 14, 15], a possible architecture for joint transmission of URLLC and eMBB types of traffic was explored. The works [16, 17] discuss the main scenarios for the implementation of URLLC and eMBB types of traffic service, technical requirements, as well as features of non-orthogonal resource sharing provided that eMBB, mMTC and URLLC consumers are located in the network. Articles [17–19] present the results of the study of the effectiveness of various modulation options for orthogonal frequency division multiplexing in the parametric space of their spectral efficiency, reliability, etc. The works [20, 21] consider options for non-orthogonal use of resources to support URLLC and eMBB connections. Among the relevant studies, we highlight the approach [22] where resource reservation is used to improve the Quality of Service for URLLC and eMBB types of traffic. In [23–25], the application of Network Slicing technology in the case of heterogeneous multiple access (both orthogonal and non-orthogonal) was studied. In addition to URLLC priority access, the authors of articles [26, 27] focus on maintaining the QoS of eMBB traffic, using methods of stochastic geometry and queuing theory.

In all the works mentioned above, the authors either view the 5G cluster as a closed system or do not take into account the features of the simultaneous operation of URLLC and eMBB technologies on the 5G platform. At the same time, the closest analogues to this research are works [28, 29]. A feature of the works [28, 29] is the use of queuing systems with random requirements for modelling URLLC-eMBB interactions. At the same time, we note that the authors of the mentioned studies try to abstract from the conditions in which the 5G cluster will operate. Thus, while increasing the universality of the results obtained, the proposed solutions lose adequacy for specific objects. We should also pay attention to the fact that the authors formulate their models [26–28] in a well-looking, but cumbersome mathematical basis, which does not allow the methods obtained to be used to analyze real 5G clusters with a large amount of communication resources. In this article, we focus on addressing these limitations.

Next, we summarize the main attributes of our research in a compact form.

The **object** of the study is the process of managing the quality of service of typified incoming requests in the open 5G ecosystem of a smart city cluster with predominant URLLC and eMBB traffic.

The **subject** of research includes methods of probability theory and mathematical statistics, recovery theory, Markov process theory, and experiment planning theory.

The **aim** of the research is formulated as follows: to create a computationally efficient, holistic concept for evaluating an arbitrary instance of the QoS policy of an ordinary 5G smart city cluster with predominant URLLC and eMBB traffic, which will make it possible to close the research gap identified as a result of the analysis of analogues.

Research **objectives** are:

- to determine the parametric space of stable and controlled characteristics that characterize the research object (presented in Section 2.1),

- to formulate a comprehensive qualitative metric for evaluating an arbitrary instance of the implementation of the research object (presented in Section 2.2),

- to formulate an accurate parameterized concept for evaluating an arbitrary instance of the QoS policy of an ordinary 5G smart city cluster (presented in Section 2.3),

- to formulate a computationally efficient approximate parameterized concept for evaluating an arbitrary instance of the QoS policy of an ordinary 5G smart city cluster (presented in Section 2.3),

- to analytically investigate partial cases of QoS policy implementation of an ordinary 5G smart city cluster in terms of an approximate evaluation concept (presented in Section 2.4),

- justify the adequacy of a mathematical apparatus proposed and demonstrate its functionality in the context of a research aim (presented in Section 3 and discussed in Section 4).

We define the **main contribution** of the research as follows. As a baseline, this research considers a QoS policy with constraints for context-defined URLLC and eMBB classes of incoming requests. Evaluating the QoS policy instance defined within the framework of the basic concept requires the formalization of both a complete qualitative metric and a computationally efficient mathematical apparatus for its calculation. The article presents accurate and approximate methods of calculating such quality parameters as the probability of loss of typed requests and the utilization ratio of the communication resource, which depend on the implementation of the estimated QoS policy. At the same time, the original parametric space includes both fixed characteristics (amount of available communication resources, load according to request classes) and controlled characteristics due to the specifics of the implementation of the basic QoS concept.

The article is organized as follows. Section 2 presents the main scientific result of the study. In particular, the research was set up, including the formation of the main parametric space of the model of the studied process. Next, a qualitative metric is formulated for evaluating an arbitrary instance of the studied process. Exact and approximate methods are formulated for the evaluation of the QoS policy model of an ordinary 5G smart city cluster (including some typical cases). In Section 3, from the standpoint of mathematical statistics and the experiment planning theory, the results of evaluating an instance of a QoS policy model of an ordinary 5G smart city cluster are presented, both using the author's approach and in comparison with an analogue. Section 4 analyzes the obtained empirical results. Section 5 draws general conclusions on the work and formulates directions for promising further research.

## 2 Materials and methods

### 2.1. Statement of the research

The focus of our study was the process of supporting 5G cluster application groups with different quality of service (QoS) requirements. Focusing on the 5G platform, three such groups are distinguished: URLLC, eMBB and mMTC. An ordinary 5G smart city cluster mainly deals with applications that belong to the first two groups (mMTC group applications are typical of industrially oriented 5G clusters). Accordingly, for an ordinary 5G cluster, an actual problem is the evaluation of QoS policies aimed at supporting URLLC and eMBB applications.

Suppose that the channel resource of an ordinary 5G cluster includes $C \in \mathbb{N}$ units, which are oriented to support Poisson flows of:

- a new incoming requests of URLLC type with intensity $\eta_u$;

- a new incoming requests of eMBB type with intensity $\eta_e$;

- a handover of incoming requests of URLLC type with intensity $\eta_{hu}$;

- a handover of incoming requests of eMBB type with intensity $\eta_{he}$;

One channel resource unit is sufficient to support a URLLC request, while $a \leq C$ channel resource units are required to support an eMBB request. To simplify the model, let's resort to the admissible assumption that eMBB requests are inelastic (at the moment of completion of the service of such a request, all resource units allocated for its support are released at the same time).

All functions that characterize the distribution of time spent on supporting {$u,e,hu,he$}-requests belong to the exponential type but with different means, which are determined by parameters {$\mu_u,\mu_e,\mu_{hu},\mu_{he}$}, respectively.

Let's define the QoS policy considering that the class of URLLC-requests has a higher priority than a class of eMBB-requests, and within the corresponding class, handover requests have a higher priority than new ones. We will ensure the balance of the QoS policy under the following conditions:

- the number of accepted $u$-requests cannot exceed the threshold $T_u$,

- if, upon receipt of a $hu$-request, the base station has at least one free resource unit at its disposal, it is directed to support this request,

- the number of accepted $e$-requests cannot exceed the threshold $T_e$,

- the number of accepted $he$-requests cannot exceed the threshold $T_{he}$,

- $0 < T_e \leq T_{he} \leq T_u \leq C$.

We will consider the ratio of channel resource utilization $\bar{C}$ and the probability of loss of requests of each of the defined types: {$P_u,P_{hu},P_e,P_{he}$} as an analytical metric of QoS policy quality.

### 2.2. Accurate estimation of the QoS policy model of an ordinary 5G smart city cluster

Suppose that $a = 1$. In the context of a research statement, at an arbitrary moment, the state of an ordinary 5G cluster is characterized by the vector $\overrightarrow{n} = (n_{\Sigma u}, n_{\Sigma e})$, where $n_{\Sigma u} = n_u + n_{hu}$ a total number of received URLLC requests and $n_{\Sigma e} = n_e + n_{he}$ a total number of received eMBB requests, respectively. A phase space of a Markov chain, which corresponds to the two-

dimensional vector $\to n$, is described by the expression

$$F := \{ n : n_{\Sigma u} = 0, \bar{C}; n_{\Sigma e} = 0, \bar{T}_{he}; n_{\Sigma u} + n_{\Sigma e} \leq C \}. \tag{1}$$

Note that an inscription of the type $y := x$ is interpreted so that the value of $y$ is determined by the expression $x$.

Taking into account the conditions for defining the QoS policy formulated in Section 2.1, elements of a Markov chain (1) generating matrix will be determined from the following ratios:

$$q(\to n, \to n') = \begin{cases} \eta_{\Sigma u} \forall \eta_{\Sigma u} \leq T_u - 1, n' = n + v_u, \\ \eta_{hu} \forall T_u \leq \eta_u \leq C - 1, n' = n + v_u, \\ \eta_{\Sigma e} \forall n_{\Sigma e} \leq T_e - 1, n' = n + v_e, \\ \eta_{he} \forall T_e \leq \eta_{he} \leq T_{he} - 1, n' = n + v_e, \\ \mu_{\Sigma u} n_{\Sigma u} \forall n' = n + v_u, \\ \mu_{\Sigma e} n_{\Sigma e} \forall n' = n + v_e, \\ 0 \forall \text{else}, \end{cases} \tag{2}$$

where $\eta_{\Sigma u} = \eta_u + \eta_{hu}$, $\eta_{\Sigma e} = \eta_e + \eta_{he}$, $\mu_{\Sigma u} = \mu_u + \mu_{hu}$, $\mu_{\Sigma e} = \mu_e + \mu_{he}$, $v_u = (0.1)$, $v_e = (0.1)$.

The analytical representation of QoS conditions embodied in expression (2) is determined through a stationary probability distribution of a process (1) states. Assume that this is a stationary probability of losing requests of a type $r$, $r \in \{u,e,hu,he\}$. According to the PASTA theorem [30], these parameters are defined as a corresponding marginal distribution of a Markov chain (1), i.e.:

$$P_u := \sum_{\to n \in F} p(\to n) I(n_u \geq T_u), \tag{3}$$

$$P_{hu} := \sum_{\to n \in F} p(\to n) \delta(n_{\Sigma u} + n_{\Sigma e}, C), \tag{4}$$

$$P_e := \sum_{\to n \in F} p(\to n) I(n_{\Sigma e} \geq T_e) + \sum_{\to n \in F} p(\to n) \delta(n_{\Sigma u} + n_{\Sigma e}, C) I(n_{\Sigma e} < T_e), \tag{5}$$

$$P_{he} := \sum_{\to n \in F} p(\to n) \delta(n_{\Sigma e}, T_{he}) + \sum_{\to n \in F} p(\to n) \delta(n_{\Sigma u} + n_{\Sigma e}, C) I(n_{\Sigma e} < T_{he}), \tag{6}$$

where $p(\to n)$ is a state $\to n \in F$ stationary probability; $\delta(\cdot)$ is the Kronecker delta; $I(E)$ is an event $E$ indicator function. Based on this terminology, the average number of used communication resources of a 5G cluster can be defined as

$$\bar{C} := \sum_{i=1}^{C} i \xi(i) = \sum_{i=1}^{C} \left( i \sum_{\to n \in F} p(\to n) \delta(n_{\Sigma u} + n_{\Sigma e}, i) \right). \tag{7}$$

Based on the Kolmogorov-Arnold theorem [31], it can be stated that the two-dimensional Markov chain (1) is characterized by reversibility. That is, in the process $F$ there is zero circulation between the states (a local balance condition is fulfilled). In turn, this allows expressing a state $(i,j)$ probability in terms of a state $(0,0)$ probability by choosing an arbitrary route between these two states in a corresponding graph. Therefore, in a case $T_{he} + T_u \leq C$ of the route $(0,0), (1,0), \ldots, (i,0), (i,1), \ldots, (i,j)$, the multiplicative solution for a stationary probability

distribution $p(\to n)$ of a process $F$ is defined by an expression

$$
p(n_{\Sigma u}, n_{\Sigma e}) = \begin{cases}
\dfrac{w_{\Sigma e}^{n_{\Sigma e}}}{n_{\Sigma e}!} \dfrac{w_{\Sigma u}^{n_{\Sigma u}}}{n_{\Sigma u}!} p(0,0) \forall n_{\Sigma e} \le T_e, n_{\Sigma u} \le T_u; \\[2ex]
\dfrac{w_{\Sigma e}^{n_{\Sigma e}}}{n_{\Sigma e}!} \dfrac{w_{hu}^{n_{\Sigma u}}}{n_{\Sigma u}!} \left(\dfrac{w_{\Sigma u}}{w_{hu}}\right)^{T_u} p(0,0) \forall \\[2ex]
\forall n_{\Sigma e} \le T_e, T_u \le n_{\Sigma u} \le C; \\[2ex]
\dfrac{w_{he}^{n_{\Sigma e}}}{n_{\Sigma e}!} \dfrac{w_{\Sigma u}^{n_{\Sigma u}}}{n_{\Sigma u}!} \left(\dfrac{w_{\Sigma e}}{w_{he}}\right)^{T_e} p(0,0) \forall \\[2ex]
\forall T_e \le n_{\Sigma e} \le T_{he}, n_{\Sigma u} \le T_u; \\[2ex]
\dfrac{w_{he}^{n_{\Sigma e}}}{n_{\Sigma e}!} \dfrac{w_{hu}^{n_{\Sigma u}}}{n_{\Sigma u}!} \left(\dfrac{w_{\Sigma e}}{w_{he}}\right)^{T_e} \left(\dfrac{w_{\Sigma u}}{w_{hu}}\right)^{T_u} p(0,0) \forall \\[2ex]
\forall T_e \le n_{\Sigma e} \le T_{he}, T_u \le n_{\Sigma u} \le C,
\end{cases}
\tag{8}
$$

where $w_{\Sigma e} := \eta_{\Sigma e}/\mu_{\Sigma e}$, $w_{\Sigma u} := \eta_{\Sigma u}/\mu_{\Sigma u}$, $w_{he} := \eta_{he}/\mu_{\Sigma e}$, $w_{hu} := \eta_{hu}/\mu_{\Sigma u}$; $p(0,0)$ is determined as a result of the normalization of

$$
p(0,0) = 1 \Big/ \left( \sum_{\to n \in F_1} \frac{w_{\Sigma e}^{n_{\Sigma e}}}{n_{\Sigma e}!} \frac{w_{\Sigma u}^{n_{\Sigma u}}}{n_{\Sigma u}!} + \left(\frac{w_{\Sigma u}}{w_{hu}}\right)^{T_u} \sum_{\to n \in F_2} \frac{w_{\Sigma e}^{n_{\Sigma e}}}{n_{\Sigma e}!} \frac{w_{hu}^{n_{\Sigma u}}}{n_{\Sigma u}!} + + \left(\frac{w_{\Sigma e}}{w_{he}}\right)^{T_e} \sum_{\to n \in F_3} \frac{w_{he}^{n_{\Sigma e}}}{n_{\Sigma e}!} \frac{w_{\Sigma u}^{n_{\Sigma u}}}{n_{\Sigma u}!} + \left(\frac{w_{\Sigma e}}{w_{he}}\right)^{T_e} \left(\frac{w_{\Sigma u}}{w_{hu}}\right)^{T_u} \sum_{\to n \in F_4} \frac{w_{he}^{n_{\Sigma e}}}{n_{\Sigma e}!} \frac{w_{hu}^{n_{\Sigma u}}}{n_{\Sigma u}!} \right),
$$

where

$$
F_1 := \{\to n \in F : n_{\Sigma e} \le T_e, n_{\Sigma u} \le T_u\},
$$

$$
F_2 := \{\to n \in F : n_{\Sigma e} \le T_e, T_u + 1 \le n_{\Sigma u} \le C\},
$$

$$
F_3 := \{\to n \in F : T_e + 1 \le n_{\Sigma e} \le T_{he}, n_{\Sigma u} \le T_u\},
$$

$$
F_4 := \{\to n \in F : T_e + 1 \le n_{\Sigma e} \le T_{he}, T_u + 1 \le n_{\Sigma u} \le C\}.
$$

If $T_{he} + T_u > C$, then a states stationary probability distribution of a process $F$ will take a form

$$
p(n_{\Sigma u}, n_{\Sigma e}) = \begin{cases}
\dfrac{w_{\Sigma e}^{n_{\Sigma e}}}{n_{\Sigma e}!} \dfrac{w_{\Sigma u}^{n_{\Sigma u}}}{n_{\Sigma u}!} p(0,0) \forall \\[2ex]
\forall 0 \le n_{\Sigma e} \le T_e, 0 \le n_{\Sigma u} \le T_u; \\[2ex]
\dfrac{w_{he}^{n_{\Sigma e}}}{n_{\Sigma e}!} \dfrac{w_{\Sigma u}^{n_{\Sigma u}}}{n_{\Sigma u}!} \left(\dfrac{w_{\Sigma e}}{w_{he}}\right)^{T_{he}} p(0,0) \forall \\[2ex]
\forall T_e + 1 \le n_{\Sigma e} \le T_{he}, 0 \le n_{\Sigma u} \le C - n_{\Sigma e}; \\[2ex]
\dfrac{w_{\Sigma e}^{n_{\Sigma e}}}{n_{\Sigma e}!} \dfrac{w_{hu}^{n_{\Sigma u}}}{n_{\Sigma u}!} \left(\dfrac{w_{\Sigma u}}{w_{hu}}\right)^{T_u} p(0,0) \forall \\[2ex]
\forall 0 \le n_{\Sigma e} \le C - T_u - 1, T_u + 1 \le n_{\Sigma u} \le C,
\end{cases}
\tag{9}
$$

where $p(0,0)$ is determined as a result of

$$
p(0,0) = 1 \Big/ \left( \sum_{\to n \in F'_1} \frac{w_{\Sigma e}^{n_{\Sigma e}}}{n_{\Sigma e}!} \frac{w_{\Sigma u}^{n_{\Sigma u}}}{n_{\Sigma u}!} + \left(\frac{w_{\Sigma e}}{w_{he}}\right)^{T_e} \sum_{\to n \in F'_3} \frac{w_{he}^{n_{\Sigma e}}}{n_{\Sigma e}!} \frac{w_{\Sigma u}^{n_{\Sigma u}}}{n_{\Sigma u}!} + \left(\frac{w_{\Sigma u}}{w_{hu}}\right)^{T_u} \sum_{\to n \in F'_4} \frac{w_{\Sigma e}^{n_{\Sigma e}}}{n_{\Sigma e}!} \frac{w_{hu}^{n_{\Sigma u}}}{n_{\Sigma u}!} \right)
$$

normalization, where

$$F'_1 := \{\rightarrow n \in F : 0 \leq n_{\Sigma e} \leq T_e, 0 \leq n_{\Sigma u} \leq T_u\},$$

$$F'_2 := \{\rightarrow n \in F : T_e + 1 \leq n_{\Sigma e} \leq T_{he}, 0 \leq n_{\Sigma u} \leq C - n_{\Sigma e}\},$$

$$F'_3 := \{\rightarrow n \in F : 0 \leq n_{\Sigma e} \leq C - T_u - 1, T_u + 1 \leq n_{\Sigma u} \leq C\}.$$

With a stationary probability distribution of target process $F$ states calculated according to expressions (8) or (9), the desired quality indicators of the formed QoS policy are calculated according to expressions (3)–(7).

The proposed approach to evaluating the QoS policy of an ordinary 5G smart city cluster is accurate. The use of this term is correct, because to calculate $p(\rightarrow n)$, $\rightarrow n \in F$, with the help of a mathematical apparatus presented in this section, researchers will have to generate the entire phase space of the states of the evaluated model. At the same time, for a large value of $C$, researchers will have difficulties with the calculations of expressions (8) or (9), because they will have to calculate a large number factorials and also raise to a power either values close to zero (at a low load on the 5G cluster) or large values (when the 5G cluster is heavily loaded). This circumstance prompted the authors to formalize the concept of the approximate calculation of the metric (3)–(7), which is presented in the following subsections of Section 2.

## 2.3. Approximate evaluation of the QoS policy model of an ordinary 5G smart city cluster

Let's formulate the concept of approximate calculation of the metric (3)–(7), for the implementation of which it will not be necessary to generate an entire estimated QoS model states phase space. An initial postulate for the concept explained below is that tabulated values are used in expressions (8), and (9). With a large difference between the values of the parameters of different types of traffic, this circumstance will allow us to achieve the desired simplification of the calculation of stationary probabilities of the states of the evaluated model with high accuracy.

First, let's assume that for the evaluated model, URLLC traffic characteristics prevail over eMBB traffic characteristics: $\mu_{\Sigma u} >> \mu_{\Sigma e}$. At the same time, we take into account that URLLC connections are short-lived compared to eMBB connections [2], however, the share of active URLLC connections in the traffic of the studied 5G cluster is dominant.

Let's divide the state graph of a investigated model by a value of a state vector's first component:

$$F = \bigcup_{i=0}^{T_{he}} F_i, \ F_i \cap F_{i'} = \emptyset, \ i \neq i', \tag{10}$$

where $F_i := \{\rightarrow n \in F : n_{\Sigma e} = i\}$. For the classes formed as a result of the implementation of partition (10) to an original phase space of a model $F$ states, it is characteristic that a transitions probabilities between states belonging to a same class significantly exceed a transitions probabilities between states belonging to different classes.

We cluster a states $F_i$ classes into the integral state $\langle i \rangle$. The corresponding clustering function is described by an expression $K(\rightarrow n)\langle i \rangle \forall \rightarrow n \in F_i, i = 0, \overline{T}_{he}$. Such a clustering function defines an integral model, which is described by a one-dimensional Markov chain with state space $\tilde{F} := \{\langle i \rangle : i = 0, \overline{T}_{he}\}$. Accordingly, the states stationary probability distribution is defined as

$$p(i,j) \approx \alpha_i(j)\beta(\langle i \rangle), \ i = 0, \overline{T}_{he}, \ j = 0, \bar{C} - i, \tag{11}$$

where $\{\alpha_i(j) : (i,j) \in F_i\}$ is the stationary probability distribution of a states within the class $S_i$,

and where $\{\beta(\langle i \rangle) : \langle i \rangle \in \tilde{F}\}$ is a stationary probability distribution of an integral model $\tilde{F}$ states. When determining these stationary state probability distributions, we will consider such cases as $h1 : T_{he} + T_u \leq C$ and $h2 : T_{he} + T_u > C$.

We denote a model $\tilde{F}$ phase matrix elements with a space of states $F_i$ as $q_i(j,l)$. For the case $h1$, generating matrix elements are defined in the same way:

$$q_i(j,l) = \begin{cases} \eta_{\Sigma u} \forall j \leq T_u - 1, l = j + 1, \\ \eta_{hu} \forall T_u \leq j \leq C - 1, l = j + 1, \\ j\mu_{\Sigma u} \forall l = j - 1, \\ 0 \forall \text{else.} \end{cases} \quad (12)$$

After analyzing expression (12), we conclude that a stationary probability distribution of a model $\tilde{F}$ states with a state space $F_i$ is identical to the stationary probability distribution of an Erlang model of type $M/M/C-i/0$ [32] states with state-dependent intensities of an arrival of new requests and an intensity $\mu_{\Sigma u}$ of serving received requests by a single device. Based on this conclusion, we write:

$$\alpha_i(j) = \begin{cases} \dfrac{w_{\Sigma u}^j}{j!} \alpha_i(0) \forall 1 \leq j \leq T_u, \\ \left(\dfrac{w_{\Sigma u}}{w_{hu}}\right)^{T_u} \dfrac{w_{hu}^j}{j!} \alpha_i(0) \forall T_u + 1 \leq j \leq C - i, \end{cases} \quad (13)$$

where

$$\alpha_i(0) = 1 \Big/ \left( \sum_{j=0}^{T_u} \frac{w_{\Sigma u}^j}{j!} + \left(\frac{w_{\Sigma u}}{w_{hu}}\right)^{T_u} \sum_{j=T_u+1}^{C-i} \frac{w_{hu}^j}{j!} \right). \quad (14)$$

Having generalized expressions (14) and (2), we formalize the relationship for calculating the generating matrix $q(\langle i \rangle, \langle i' \rangle), \langle i \rangle, \langle i' \rangle \in \tilde{F}$:

$$q(\langle i \rangle, \langle i' \rangle) = \begin{cases} \eta_{\Sigma e}(1 - \alpha_i(C - i)) \forall \\ \forall 0 \leq i \leq T_e - 1, i' = i + 1, \\ \eta_{he}(1 - \alpha_i(C - i)) \forall \\ \forall T_e \leq i \leq T_{he} - 1, i' = i + 1, \\ i\mu_e \forall i' = i - 1, \\ 0 \forall \text{else.} \end{cases} \quad (15)$$

The relations defined by expression (15) will allow us to determine a stationary probability distribution of a model $\tilde{F}$ states (a one-dimensional birth and death Markov process):

$$\beta(\langle i \rangle) = \frac{\beta(\langle 0 \rangle)}{i! \mu_{\Sigma e}^i} \prod_{j=1}^{i} q(\langle i - 1 \rangle, \langle i \rangle), \ i = 1, \overline{T}_{he}, \quad (16)$$

where

$$\beta(\langle 0 \rangle) = 1 \Big/ \left( 1 + \sum_{i=1}^{T_{he}} \frac{1}{i! \mu_{\Sigma e}^i} \prod_{j=1}^{i} q(\langle i - 1 \rangle, \langle i \rangle) \right). \quad (17)$$

Finally, having determined the characteristics (13)–(17), we formalize the expressions for the approximate estimation of the metric (3)–(7) for the evaluated instance of the QoS policy model at $\mu_{\Sigma u} >> \mu_{\Sigma e}$:

$$P_u \approx \sum_{i=0}^{T_{he}} \beta(\langle i \rangle) \sum_{j=T_u}^{C-i} \alpha_i(j), \tag{18}$$

$$P_{hu} \approx \sum_{i=0}^{T_{he}} \beta(\langle i \rangle) \alpha_i(C - i), \tag{19}$$

$$P_e \approx \sum_{i=T_e}^{T_{he}} \beta(\langle i \rangle) + \sum_{i=0}^{T_e-1} \beta(\langle i \rangle) \alpha_i(C - i), \tag{20}$$

$$P_{he} \approx \beta(\langle T_{he} \rangle) + \sum_{i=0}^{T_{he}-1} \beta(\langle i \rangle) \alpha_i(C - i), \tag{21}$$

$$\bar{C} \approx \sum_{i=1}^{C} i \sum_{j=0}^{g(i)} \beta(\langle j \rangle) \alpha_j(i - j), \tag{22}$$

where $g(i) = \begin{cases} i \forall 1 \leq i \leq T_{he}, \\ T_{he} \forall T_{he} + 1 \leq i \leq C. \end{cases}$

For a case $h2$, a stationary probability distribution of a model $\tilde{F}$ states with state space $S_i$, $i = 0, C - \bar{T}_u - 1$, is also determined by relation (15), and a stationary probability distribution of a model $\tilde{F}$ states with state space $S_i$, $i = C - \bar{T}_u, T_{he}$, is identical to a stationary probability distribution of an Erlang model $M/M/C-i/0$ states with load $w_{\Sigma u}$. Accordingly, expressions (19)–(22) obtained for a case $h1$ will remain adequate for the case $h2$. The qualitative indicator $P_u$ for the case $h2$ is characterized by the expression

$$P_u \approx \sum_{i=0}^{C-T_u} \beta(\langle i \rangle) \sum_{j=T_u}^{C-i} \alpha_i(j) + \sum_{i=C}^{T_{he}} \beta(\langle i \rangle) \alpha_i(C - i). \tag{23}$$

## 2.4. Approximate evaluation of typical QoS policy models of an ordinary 5G smart city cluster

Let's consider several typical implementations of the QoS policy of an ordinary 5G smart city cluster.

We formalize the evaluation procedure of the ***fully available*** QoS policy, the characteristic feature of which is to ignore the differences between URLLC and eMBB traffic types: $T_e = T_{he} = T_u = C$ (this corresponds to the case $h2$ described in the previous subsection). It is obvious that in such a case $P_u = P_{hu} = P_e = P_{he} = P$. Considering the PASTA theorem, let us generalize expressions (19)–(23) for the following partial case:

$$P \approx \sum_{i=0}^{C} EB(w_{\Sigma u}, C - i) \beta(\langle i \rangle),$$

$$\bar{C} \approx \sum_{i=1}^{C} i \sum_{j=0}^{i} \beta(\langle j \rangle) \alpha_i(i - j)$$

where $EB(x,y)$ is the classical expression for an Erlang model $M/M/y/0$ (Erlang B Model) at load $x$.

The stationary distribution (15) for this partial case is defined as a stationary distribution of a model $M/M/C-i/0$ states probabilities with load $w_{\Sigma u}$:

$$q(\langle i \rangle, \langle i' \rangle) = \begin{cases} \eta_{\Sigma e}(1 - EB(w_{\Sigma u}, C - i))\forall \\ \forall 0 \leq i \leq C - 1, i' = i + 1; \\ i\mu_{\Sigma e}\forall i > 0, i' = i - 1; \\ 0\forall \text{else}. \end{cases}$$

Now we formalize the QoS evaluation procedure of a ***single-threshold*** policy, the characteristic feature of which is to ignore discrepancies between new and handover requests for both URLLC and eMBB traffic types: $T_e = T_{he}$, $T_u = C$, $T_{he} < T_u$ (this corresponds to the case $h2$). Based on expressions (19)–(23) for such a partial case, we obtain:

$$P_u = P_{hu} \approx \sum_{i=0}^{T_{he}-1} \beta(\langle i \rangle) \alpha_i(C - i),$$

$$P_e = P_{he} \approx \sum_{i=0}^{T_{he}} \beta(\langle i \rangle) \alpha_i(C - i).$$

The qualitative parameter $\bar{C}$ for this partial case is calculated by expression (22). Accordingly, we present the relation (15) as follows:

$$q(\langle i \rangle, \langle i' \rangle) = \begin{cases} \eta_{\Sigma e}(1 - \alpha_i(C - i))\forall \\ \forall 0 \leq i \leq T_{he} - 1, i' = i + 1; \\ i\mu_{\Sigma e}\forall i > 0, i' = i - 1; \\ 0\forall \text{else}. \end{cases}$$

Finally, we formalize the QoS policy evaluation procedure for a 5G cluster, where ***eMBB traffic parameters dominate over URLLC traffic parameters***: $\eta_{\Sigma u} << \eta_{\Sigma e}$, $\mu_{\Sigma u} << \mu_{\Sigma e}$.

For such a partial case, an initial states $F$ phase space is divided by a value of a state vector's second component: $F = \bigcup_{i=0}^{C} F_i$, $F_i \cap F_{i'} = \emptyset$, $i \neq i'$, where $F_i := \{\to n \in F : n_{\Sigma u} = i\}$. Using the semantically identical statements presented in the previous subsection, we formulate the information technology for determining the qualitative metric (3)–(7) for the following partial case:

*Stage 1*. We calculate a stationary probability distributions of a model $\tilde{F}$ states with a state space $F_i$, $i = \overline{0, C}$. Such distributions:

- - for a model with a state space $F_i$, $i = \overline{0, C - T_{he}}$, coincide with a probability distributions of a model $M/M/T_{he}/0$ states;

- - for a model with a state space $F_i$, $i = \overline{C - E_{he} + 1, C}$, coincide with a probability distributions of a model $M/M/C-i/0$ states.

At the same time, in all models, an intensity of a unit of communication resource service is equal to $\mu_{\Sigma e}$, and an intensity of a flow of incoming requests $\eta_l$ depends on a state of $l$: $\eta_l =$

$$\begin{cases} \eta_{\Sigma e} \forall l \leq T_e - 1, \\ \eta_{he} \forall l \geq T_e. \end{cases}$$ *Stage 2.* If $T_{he} + T_u \leq C$, then an elements of an integral model generating matrix are determined by the ratio

$$q(\langle i \rangle, \langle i' \rangle) = \begin{cases} \eta_{\Sigma u} \forall i \leq T_u - 1, i' = i + 1, \\ \eta_{he} \forall T_u \leq i < C, i' = i + 1, \\ \eta_{he} \displaystyle\sum_{j=0}^{C-i-1} \alpha_i(j) \forall C - T_{he} \leq i < C, i' = i + 1, \\ i\mu_{\Sigma u} \forall i' = i - 1, \\ 0 \forall \text{else.} \end{cases}$$

If the inequality $T_{he} + T_u \leq C$ is not fulfilled, then we proceed to *Stage 3*, otherwise, we proceed to *Stage 4*.

*Stage 3.* Elements of an integral model generating matrix are determined by a ratio

$$q(\langle i \rangle, \langle i' \rangle) = \begin{cases} \eta_{\Sigma u} \forall i \leq C - T_{he} - 1, i' = k + 1, \\ \eta_{he} \displaystyle\sum_{j=0}^{C-i-1} \alpha_i(j) \forall T_e \leq i \leq T_{he} - 1, i' = k + 1, \\ i\mu_{\Sigma u} \forall i' = i - 1, \\ 0 \forall \text{else.} \end{cases}$$

*Stage 4.* Similarly to (16), and (17), we find a stationary probability distributions of a model $\beta(\langle i \rangle)$ states, $i = \overline{0, C}$, and, after determining the stationary probability distribution of a model $\tilde{F}$ by expression (11), we calculate an approximate value of an evaluated QoS policy quality indicators. In this case:

- if the inequality $T_{he} + T_u \leq C$ is satisfied, then the approximate values of the quality parameters (3)–(7) for the evaluated QoS policy are calculated using the expressions

$$P_u \approx \sum_{i=T_u}^{C} \beta(\langle i \rangle), \tag{24}$$

$$P_{hu} \approx \sum_{i=C-T_{he}}^{C} \beta(\langle i \rangle)\alpha_i(C - i), \tag{25}$$

$$P_e \approx \sum_{i=0}^{C-T_{he}} \beta(\langle i \rangle)\sum_{j=T_e}^{T_{he}} \alpha_i(j) + \sum_{i=C-T_{he}+1}^{C} \beta(\langle i \rangle)\alpha_i(C - i), \tag{26}$$

$$P_{he} \approx \sum_{i=0}^{C-T_{he}} \beta(\langle i \rangle)\alpha_i(T_{he}) + \sum_{i=C-T_{he}+1}^{C} \beta(\langle i \rangle)\alpha_i(C - i), \tag{27}$$

$$\bar{C} \approx \sum_{i=1}^{T_{he}} i \sum_{j=0}^{i} \beta(\langle i \rangle)\alpha_i(i-j) + \sum_{i=T_{he}+1}^{C} i \sum_{j=i-T_{he}}^{i} \beta(\langle i \rangle)\alpha_i(i-j), \tag{28}$$

at $\alpha_C(0) := 1$;

- if the inequality $T_{he} + T_u \leq C$ is not fulfilled, then the approximate values of the quality parameters $P_{hu}$, $P_e$, $P_{he}$, $\bar{C}$ for the evaluated QoS policy are calculated according to expressions (25)–(28), respectively, and the approximate value of the quality parameter $P_u$ is calculated according to the expression

$$P_u \approx \sum_{i=T_u}^{C} \beta(\langle i \rangle) + \sum_{i=C-T_{he}}^{T_{u-1}} \beta(\langle i \rangle)\alpha_i(C-i). \tag{29}$$

## 3 Results

Earlier, we noted that the exact method of QoS policy estimation proposed in Section 2.2 is computationally acceptable for estimating target models of moderate dimension, which is considered by the value of $C$. In contrast, the mathematical apparatus presented in Sections 2.3 and 2.4 allows for QoS evaluation of large-scale models in an almost arbitrary range of measurement of load characteristics of both URLLC and eMBB traffic. At the same time, the main thing is to establish the adequacy of the proposed approximate methods.

The value of the initial parameters for simulation modelling is set: $QoS_1$: $C = 160$, $T_u = 140$, $T_{he} = 120$, $\eta_u = 10$, $\eta_{hu} = 6$, $\eta_e = 4$, $\eta_{he} = 3$, $\mu_{\Sigma u} = 10$, $\mu_{\Sigma e} = 2$. For this initial set, we calculate the value of the qualitative metric $r = \{P_u, P_{hu}, P_e, P_{he}, \bar{C}\}$ by the accurate (Exact) (*EM* which is presented in Section 2.2) and Approximate (*AM* which is presented in Section 2.3) Methods by changing a value of a threshold $T_e$ in a range of {10,20,. . .,120}. The calculated functional dependence $r = f(T_e, \{EM, AM\})$ is visualized in Figs 1–3.

We note that with the above input set for the evaluated QoS model, the values of quality indicators $P_u$ and $P_{hu}$ practically coincide (see Fig 1). The reason for this is that the applied values for the characteristics $C$ and $T_u$ are close. Let's spread the values of these characteristics: $QoS_2$: $C = 250$, $T_u = 200$, $T_{he} = 120$, $\eta_u = 10$, $\eta_{hu} = 6$, $\eta_e = 4$, $\eta_{he} = 3$, $\mu_{\Sigma u} = 10$, $\mu_{\Sigma e} = 2$.

For the set $QoS_2$, we calculate the value of the qualitative metric $r = \{P_u, P_{hu}, P_e, P_{he}, \bar{C}\}$ by an approximate method by changing a value of a threshold $T_e$ in a range {10,20,. . .,120}. The calculated functional dependence $r = f(T_e, \{EM, AM\})$ for model $QoS_2$ is visualized in Figs 4–6. Note that for the model $QoS_2$, the absolute error between the calculations of the quality metric $r$ was so small that it is not visually distinguishable on the graphs.

Shown in Figs 1–6 results confirm an adequacy of a mathematical apparatus presented in Sections 2.2, and 2.3 because the results calculated using the Accurate and Approximate Methods coincided for different sets of input data. It remains to present the results that will confirm a functionality of a mathematical apparatus presented in Section 2 for comparing different QoS policies in the parametric space of the quality metric $r$. We compare the author's QoS policy $QoS_A$ described in section 2.1 with the known QoS policy [22] $QoS_C$, which takes into account the phenomenon of reservation of communication resources. The QoS policy [22] provides:

- if at least one channel resource unit is free when an *hu*-request incomes, then this request is accepted;

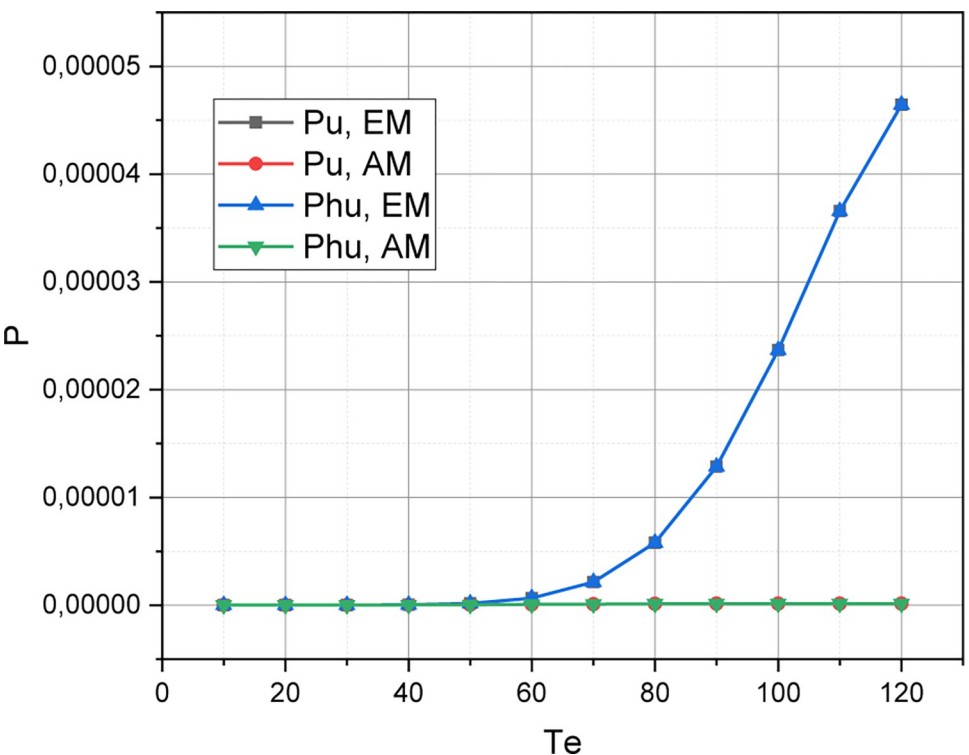

Fig 1. Visualization of dependency $\{P_u, P_{hu}\} = f(T_e, \{EM, AM\})$.

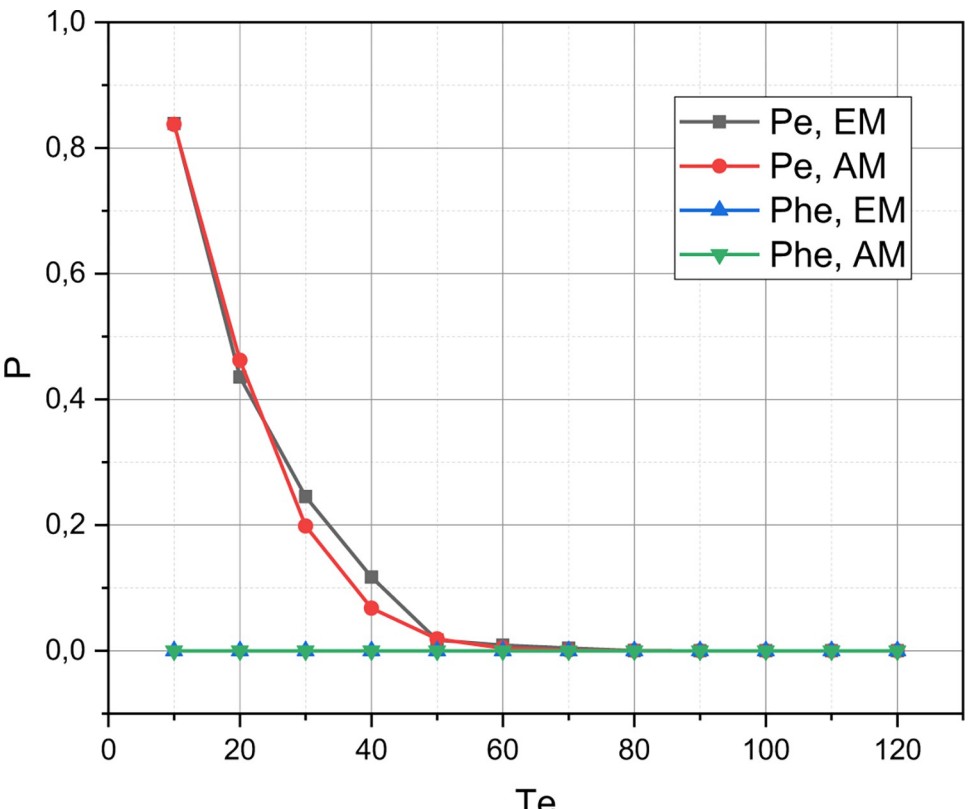

Fig 2. Visualization of dependency $\{P_e, P_{he}\} = f(T_e, \{EM, AM\})$.

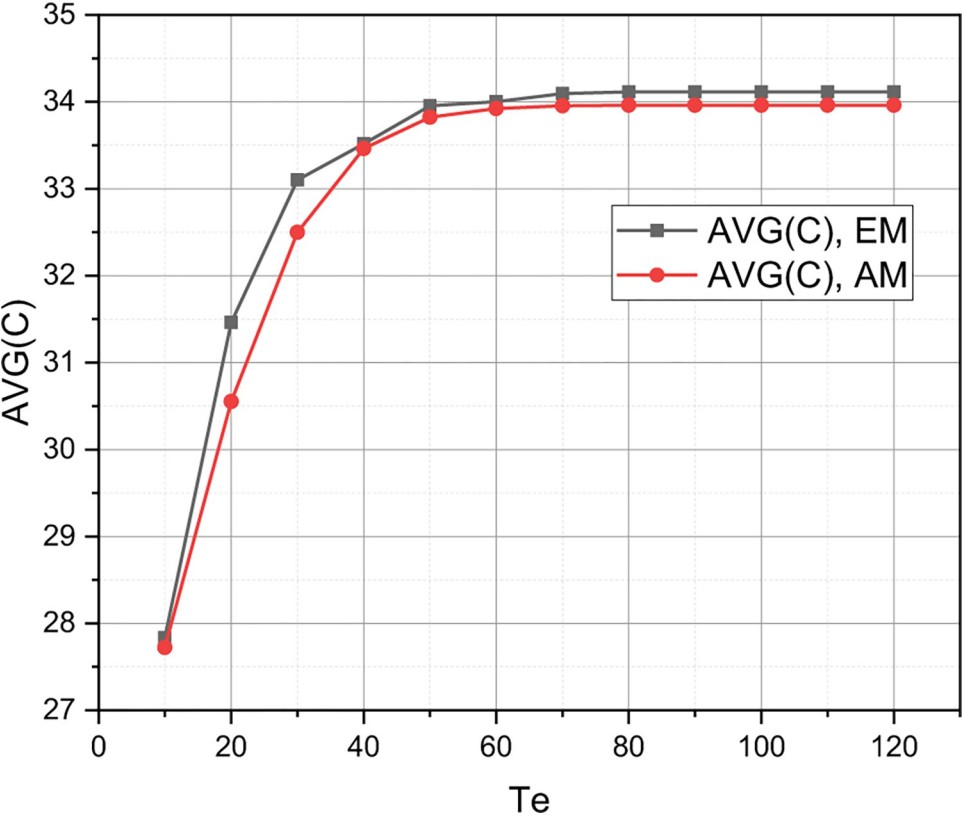

**Fig 3. Visualization of dependency $\bar{C} = f(T_e, \{EM, AM\})$.**

- an incoming $u$-request is accepted if, at a moment of its receipt, a number of occupied units of the channel resource is less than $N_u$;

- an incoming $e$-request is accepted if, at a moment of its receipt, a number of occupied units of the channel resource is less than $N_e$;

- an incoming $he$-request is accepted if, at a moment of its receipt, a number of occupied units of the channel resource is less than $N_{he}$.

In both the author's model $QoS_A$ and the competing model $QoS_C$, the total number of channel resource units $C$ is constant, and the parameters $\langle T_u, T_{he}, T_e \rangle$ (for $QoS_A$) and parameters $\langle N_u, N_{he}, N_e \rangle$ (for $QoS_C$) are controllable.

Of research interest is the comparison of policies $QoS_A$ and $QoS_C$ in the qualitative metric $r$. The rest of the input data for the experiment are defined by the set $QoS_1$ (see the beginning of this section). The results of the comparison of the competing QoS policy with reservation $QoS_C$ and the author's QoS policy $QoS_A$ are presented in Figs 6–9.

## 4 Discussion

Let's start a discussion by analyzing a simulation results, which are shown in Figs 1–3. As already emphasized, the purpose of this experiment was to prove an adequacy of the concept *AM* presented in Section 2.3 for the approximate evaluation of an arbitrary instance of the QoS policy model of an ordinary 5G smart city cluster, represented in the parametric space of the initial characteristics defined in Section 2.1. The concept *EM* for accurate estimation of a

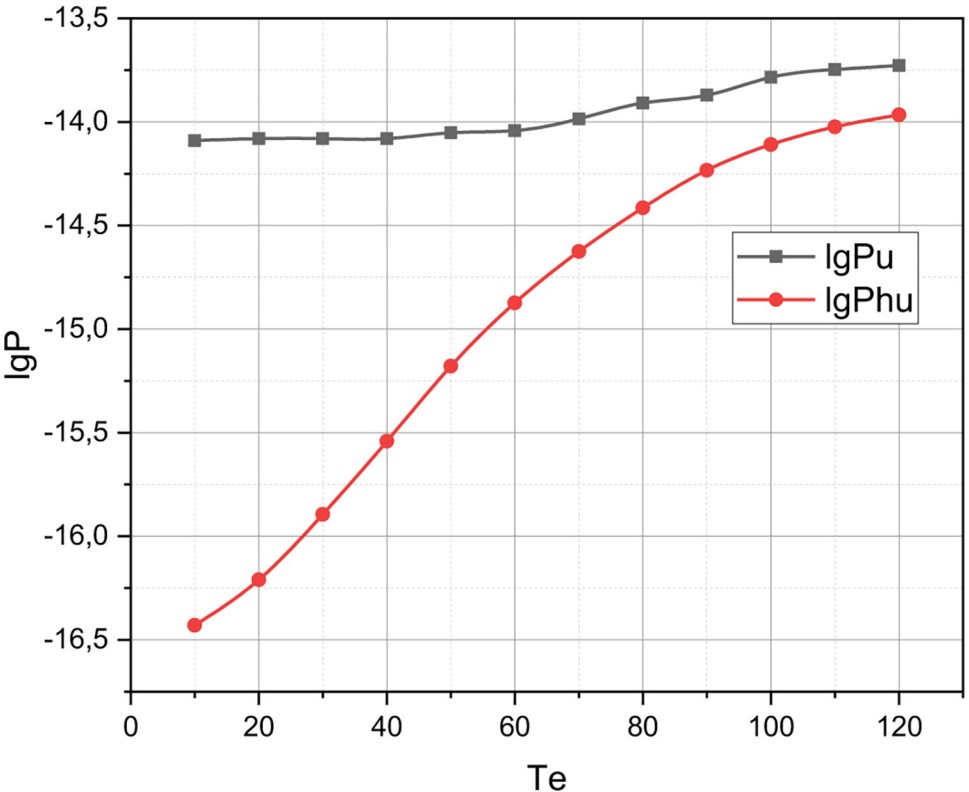

**Fig 4. Visualization of dependence $\{P_u, P_{hu}\} = f(T_e)$ for model $QoS_2$.**

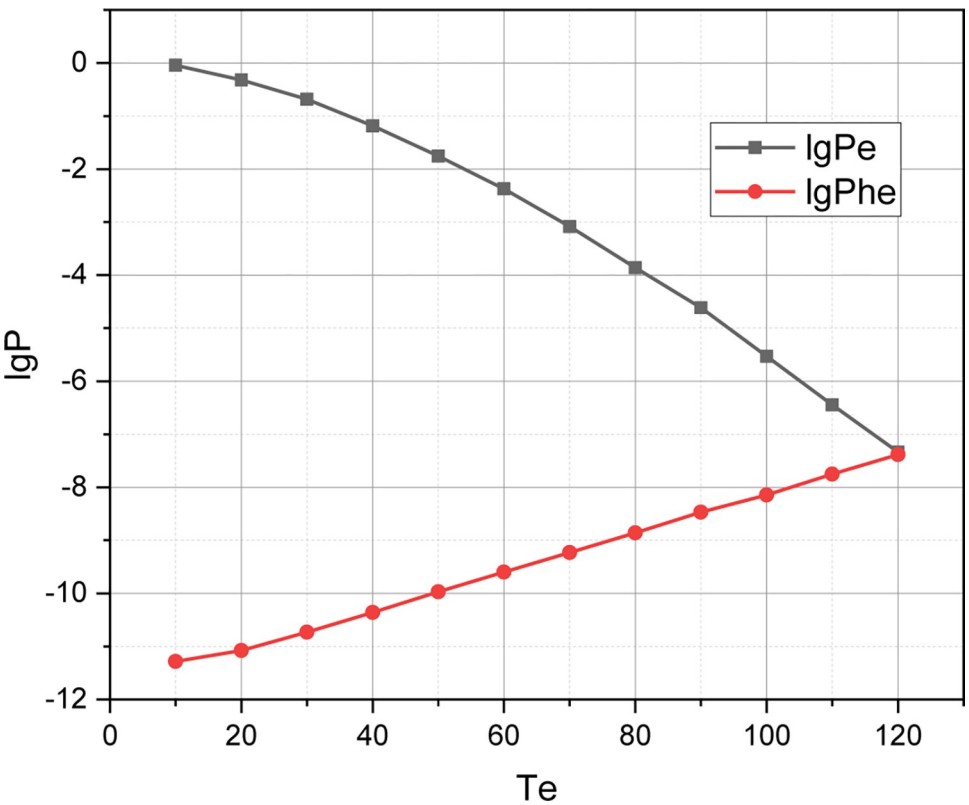

**Fig 5. Visualization of dependence $\{P_e, P_{he}\} = f(T_e)$ for model $QoS_2$.**

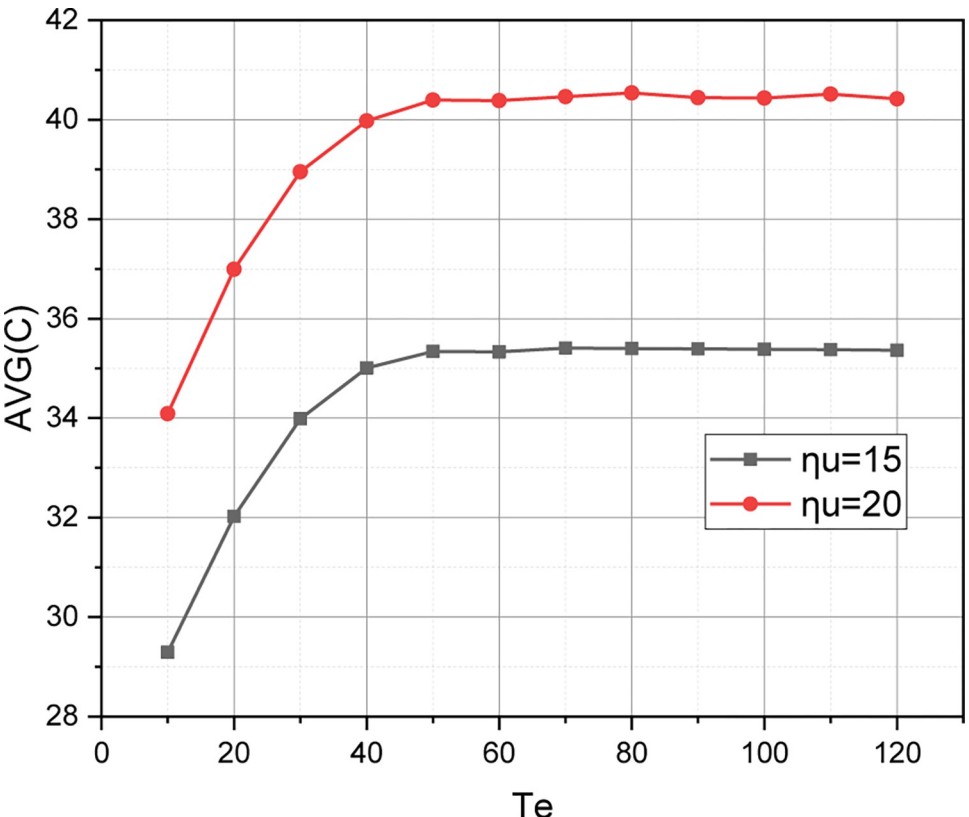

**Fig 6. Visualization of dependence $\bar{C} = f(T_e, \eta_u = \{15, 20\})$ for model $QoS_2$.**

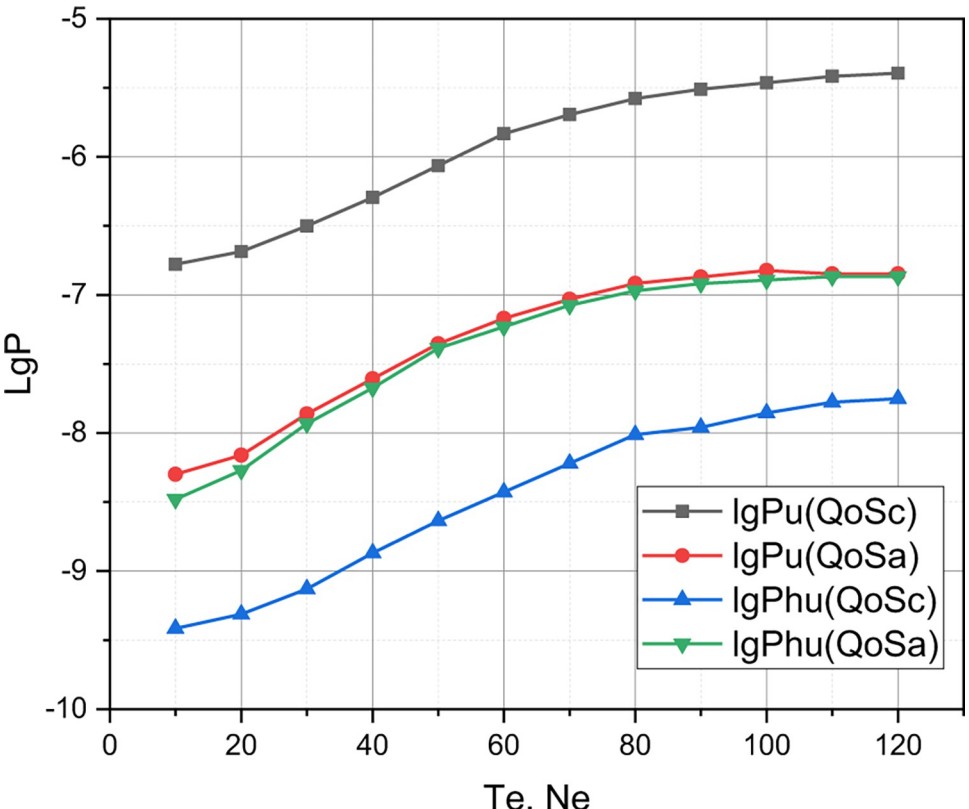

**Fig 7. Comparison of $QoS_C$ and $QoS_A$ policies in the parametric space of quality indicators $P_u$, $P_{hu}$.**

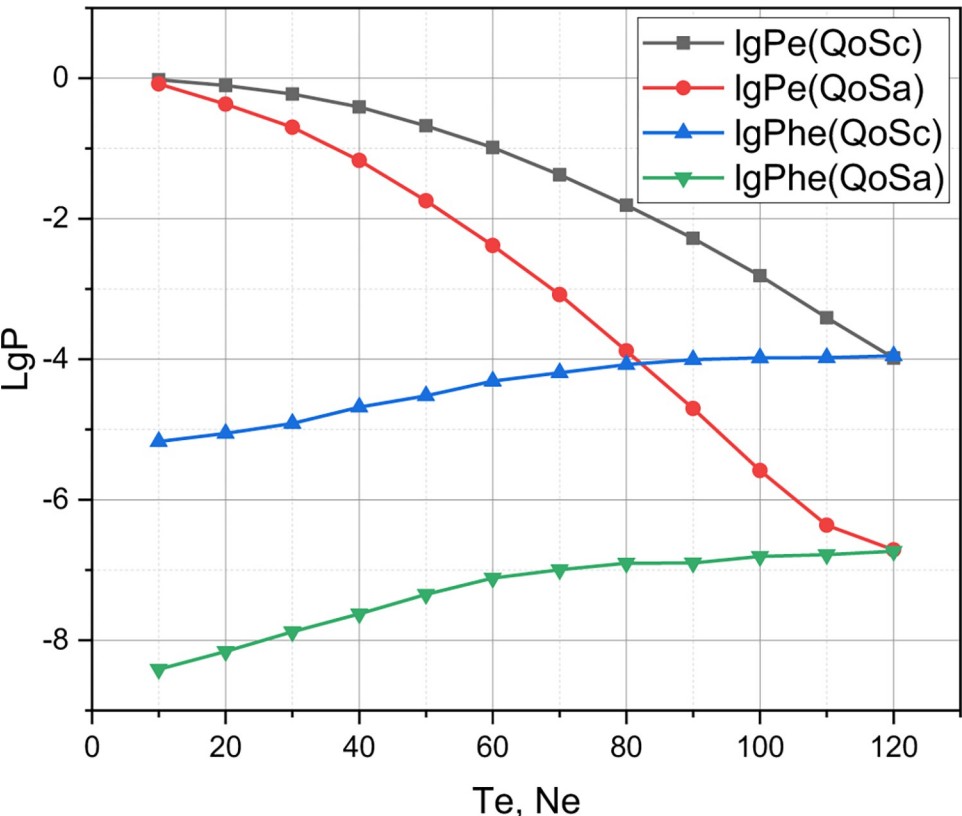

**Fig 8. Comparison of $QoS_C$ and $QoS_A$ policies in the parametric space of quality indicators $P_e$, $P_{he}$.**

QoS model of the same type, presented in Section 2.2, served as an etalon for comparison. From Figs 1–3 it can be seen that the absolute error inherent in the concept $AM$ is low enough to recognize a mathematical apparatus presented in Section 2.3 as adequate for profile use. The discrepancy between the $r$-estimates calculated using $EM$ and $AM$ is particularly small at low threshold values of $T_e$. Interestingly, this discrepancy reaches its highest values at an average value of a threshold $T_e$. The explanation for this fact is that the accuracy of the concept $AM$ is determined not so much by a values of an initial characteristics of the evaluated instance of a QoS model, but by the ratio of these characteristics: $w_{\Sigma e} := \eta_{\Sigma e}/\eta_{\Sigma e}$, $w_{\Sigma u} := \eta_{\Sigma u}/\mu_{\Sigma u}$, $w_{he} := \eta_{he}/\mu_{\Sigma e}$, $w_{hu} := \eta_{hu}/\mu_{\Sigma u}$. It is for the abscissa range $T_e \in [20,50]$ that these relations acquired the smallest values. As the ratio $\eta_{\Sigma u}/\eta_{\Sigma e}$, $\mu_{\Sigma u}/\mu_{\Sigma e}$ increases, so does the accuracy of the concept $AM$. By the way, we chose the parameter $T_e$ as the basis for demonstrating the research results because it is eMBB requests that are both the most frequent and the most resource-intensive in the traffic of an ordinary 5G smart city cluster. It should also be noted that the concept $AM$ demonstrates high accuracy for loads characterized by unevenness $\eta_{\Sigma u} << \eta_{\Sigma e}$, $\mu_{\Sigma u} << \mu_{\Sigma e}$ (in this case, the corresponding version $AM$ is used–see Section 2.4).

Presented in Figs 4–6 results were, first of all, aimed at demonstrating the sensitivity of the quality metric $r$ to the initial characteristics of the evaluated QoS model (the graphs shown in these figures represent the evaluation of instances $QoS_1$ and $QoS_2$ in the metric $r$). From the obtained results, it can be seen that an increase within a permissible limits of a value of one of the thresholds $\{T_u, T_e, T_{he}\}$ has a positive effect only on the value of the qualitative indicator, which determines the probability of losing a request of the corresponding type. An increase in the value of the threshold $T_u$ in model $QoS_2$ compared to model $QoS_1$ led to a decrease in the

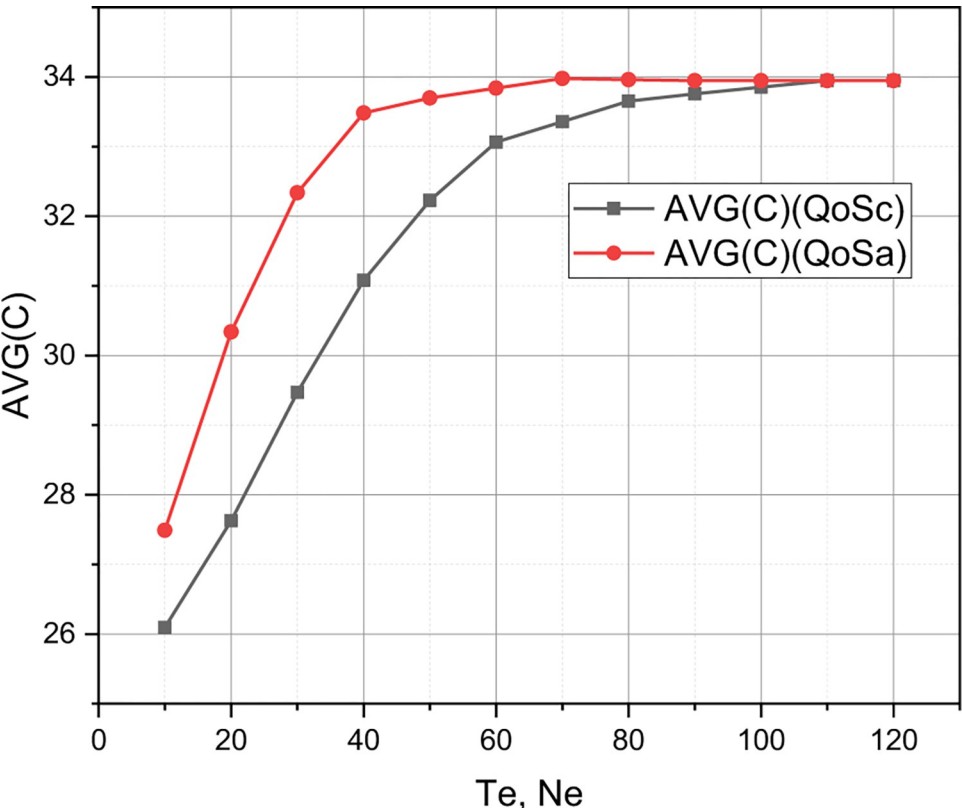

**Fig 9. Comparison of $QoS_C$ and $QoS_A$ policies in the parametric space of quality indicator $\bar{C}$.**

value of $P_u(QoS_2)$ with a simultaneous increase in the values of $P_{hu}(QoS_2)$, $P_e(QoS_2)$, $P_{he}(QoS_2)$. It can also be seen that an increase in a value of the initial characteristic $C$ in model $QoS_2$ compared to model $QoS_1$ led to an increase in the coefficient $\bar{C} \in r$ (see Fig 6). Separately, we note (see Fig 4) that the growth of the threshold $T_e$ has a positive effect on the dynamics of the value of the quality indicator $P_{hu}$ for model $QoS_2$, while the value of the quality indicator $P_u$ is practically independent of the change in the value of the threshold $T_e$. However, this circumstance is reinforced by the fact that for modelling $QoS_2$ the threshold $T_u = 200$, and the value of the threshold $T_e \in [10,120]$, that is, the studied 5G cluster is more oriented towards serving URLLC traffic. Fig 5 shows that as the value of the threshold $T_e$ increases, the values of the quality indicators $P_e$, $P_{he}$ go towards each other. At the same time, the value of the indicator $P_e$ shows negative dynamics, and the value of the indicator $P_{he}$ shows positive dynamics. Therefore, as the threshold $T_e$ increases, the 5G cluster with the model $QoS_2$ becomes more of a repeater for the handover of eMBB traffic.

Finally, let's comment on the presented in Figs 6–9 results of comparison of the competing QoS policy with reservation $QoS_C$ and the author's QoS policy $QoS_A$ for the 5G cluster, the initial parameters of which were set by the set $QoS_1$. From Figs 7 and 8, it can be seen that the graphs that correspond to the qualitative indicators $P_u$, $P_e$, $P_{he}$ for the model $QoS_A$ are superior to the competitor $QoS_C$, while the graphs that correspond to the qualitative indicator $\bar{C}$ (see Fig 9) for these competing models practically coincide. Therefore, the model $QoS_C$ is superior to the author's model only in terms of quality indicators $P_{hu}$. Taking into account that the complexity of the implementation of both models $QoS_C$ and $QoS_A$ is the same, it can be reasonably argued that for a 5G cluster, the initial characteristics of which are caused by the set $QoS_1$,

the author's model outperformed the competitor. However, we note the locality of this conclusion, because for a different set of values of the initial characteristics, the evaluation results of these QoS models may differ significantly, which requires a separate investigation.

## 5 Conclusions

A typical element of the smart city's information and communication space is a 5G cluster, which is focused on serving both new and handover requests because it is an open system. In an ordinary 5G smart city cluster, URLLC and eMBB traffic types prevail. The formation of an effective QoS policy for such an object (taking into account the potentially active slicing technology) is an urgent task.

As a baseline, this research considers a QoS policy with constraints for context-defined URLLC and eMBB classes of incoming requests. Evaluating the QoS policy instance defined within the framework of the basic concept requires the formalization of both a complete qualitative metric and a computationally efficient mathematical apparatus for its calculation. The article presents accurate and approximate methods of calculating such quality parameters as the probability of loss of typed requests and the utilization ratio of the communication resource, which depend on the implementation of the estimated QoS policy. At the same time, the original parametric space includes both fixed characteristics (amount of available communication resources, load according to request classes) and controlled characteristics due to the specifics of the implementation of the basic QoS concept.

The paper empirically proves the adequacy of the presented mathematical apparatus for evaluating the QoS policy defined within the scope of the research. Also, in the proposed qualitative metric, a comparison of the author's concept with a parametrically close analogue, determined taking into account the reservation of communication resources, was made. The results of the comparison testify in favour of the superiority of the author's approach.

Note that the presented mathematical apparatus has the potential to expand the range of QoS policies admissible for evaluation. The field of potentially acceptable objects includes network technologies that provide for the formation of queues for incoming requests. The formalization of relevant analytical adaptations of the mathematical apparatus presented in the article is a promising direction for **further research**.

Also note that we presented the main scientific result of the study based on mathematical statistics, and recovery theory. This approach is characterized by sufficient universality (subject to adequacy control) and low computational complexity. The shortcoming of the proposed approach is that the procedure for adapting the proposed mathematical apparatus to the specific conditions of 5G network deployment cannot be automated. In this context, a promising direction for further research is also the synthesis of models based on artificial intelligence, which can complement the result presented in the article. The authors see works such as [33–37] as analogues for this area of research.

## Supporting information

**S1 Data.**
(XLS)

## Acknowledgments

The authors are grateful to all persons and organizations that contributed to the publication of the article.

## Author Contributions

**Conceptualization:** Viacheslav Kovtun.

**Data curation:** Krzysztof Grochla, Torki Altameem, Mohammed Al-Maitah.

**Formal analysis:** Viacheslav Kovtun.

**Funding acquisition:** Torki Altameem.

**Investigation:** Viacheslav Kovtun.

**Methodology:** Viacheslav Kovtun.

**Project administration:** Viacheslav Kovtun.

**Resources:** Krzysztof Grochla, Torki Altameem, Mohammed Al-Maitah.

**Software:** Viacheslav Kovtun.

**Supervision:** Viacheslav Kovtun.

**Validation:** Krzysztof Grochla, Torki Altameem, Mohammed Al-Maitah.

**Visualization:** Viacheslav Kovtun.

**Writing – original draft:** Viacheslav Kovtun.

**Writing – review & editing:** Viacheslav Kovtun.

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
