## [Decision Letter · Decision Letter 0]

9 Nov 2023

PONE-D-23-33802Evaluation of the QoS Policy Model of an Ordinary 5G Smart City Cluster with Predominant URLLC and eMBB TrafficPLOS ONE

Dear Dr. Kovtun,

Thank you for submitting your manuscript to PLOS ONE. After careful consideration, we feel that it has merit but does not fully meet PLOS ONE’s publication criteria as it currently stands. Therefore, we invite you to submit a revised version of the manuscript that addresses the points raised during the review process.

We look forward to receiving your revised manuscript.

Kind regards,

Vincent Omollo Nyangaresi, Ph.D

Academic Editor

PLOS ONE

Journal Requirements:

"The authors are grateful to King Saud University, Saudi

Arabia for funding this work through Researchers Supporting Project number

(RSP2023R503), and also grateful to all persons and organizations that contributed

to the publication of the article."

"T. A.,

RSP2023R503,

Torki Altameem,

The funder had a role in data collection."

Additional Editor Comments:

1) All acronyms must be written in full the first time they are used within text

2)"...The results of the comparison testify in favour of the superiority of the author's approach..."

Give a summary of the obtained results to support the above claim.

3) Just before the "2 Materials and Methods" section, give an outline on how this paper is organized.

4)"...The focus of our study will be the process..."

Change the above to "..The focus of our study was the process..."

5) Ensure that all the presented figures are appropriately interpreted and discussed.

Reviewers' comments:

Reviewer's Responses to Questions

**Comments to the Author**

1. Is the manuscript technically sound, and do the data support the conclusions?

Reviewer #1: Yes

Reviewer #2: Yes

2. Has the statistical analysis been performed appropriately and rigorously? 

Reviewer #1: Yes

Reviewer #2: Yes

3. Have the authors made all data underlying the findings in their manuscript fully available?

Reviewer #1: Yes

Reviewer #2: Yes

4. Is the manuscript presented in an intelligible fashion and written in standard English?

Reviewer #1: Yes

Reviewer #2: Yes

5. Review Comments to the Author

Reviewer #1: 1- The abstract did not mention the adopted standards or their numerical values that confirm the superiority of the proposed approach over other methods.

2- Unify the format of keywords by writing them all with the first capital letter

3- In the introduction, the researcher does not explain what failures previous research has suffered from, or how the research presented here excelled in relation to them. In addition, the structure of the research was not explained in a paragraph at the end of the introduction

4- Standardize the format of references

Reviewer #2: Authors proposed efficient mathematical apparatus for evaluating the QoS. Here are some comments:

1- What is the problem to be solved and the existing challenges should be explained in the abstract.

2- The key comparison values should be mentioned in the abstract. These values show the efficiency of the proposed method.

3- In introduction, “URLLC technology is designed to support mission-critical communications with low latency, high security and reliability. Finally, eMBB technology aims to provide high data transfer rates.”, please add proof to this claim.

4- In introduction, “To a first approximation, this space can be segmented into industrial, specialized and civil clusters that require wireless communication coverage.”, What was this segmentation based on?

5- In introduction, “At the same time, typical information exchange in such a 5G cluster requires the implementation of URLLC (ensuring the information needs of critical infrastructure and institutions) and eMBB (ensuring the information needs of citizens and the service sector) technologies.”, Please add a reference to support this statement.

6- Adding a recent reference that addresses the challenges of QoS in the introduction and related work,

The “Related Work” section lacks of enough references. I strongly recommend that the author improve this section by adding references that support all the claims, the challenges of QoS, and motivation of the problem. The author may precisely and comprehensively point out the current issues and existing solutions. I suggest adding more related reference such as:

https://ieeexplore.ieee.org/abstract/document/9799997

https://link.springer.com/chapter/10.1007/978-981-19-1653-3_9

https://link.springer.com/chapter/10.1007/978-3-031-19523-5_6

https://ieeexplore.ieee.org/abstract/document/9768338

https://link.springer.com/chapter/10.1007/978-3-030-95987-6_1

7- In introduction, “We should also pay attention to the fact that the authors formulate their models in a well-looking, but cumbersome mathematical basis “, the mathematical foundations must be mentioned.

8- The article did not address the research gap.

9- It also did not explain sufficiently how to achieve the objectives.

10- The symbols mentioned in the statement of research are not defined.

11- In statement of research, “We will ensure the balance of the QoS policy under the following conditions:”, What was relied upon in presenting these conditions?

12- In 2.2, “The proposed approach to evaluating the QoS policy of an ordinary 5G smart city cluster is accurate.”, What is the proof?

13- In 2.3, “At the same time, we take into account that URLLC connections are short-lived compared to eMBB connections”, please add reference as proof.

14- In 2.3, “After analyzing expression (12), we conclude that a stationary probability distribution of a model F states with a state space i F is identical”, the explanation must be detailed regarding identical.

15- In 2.4, What is the effect of C on QoS?

16- In 2.4, “If the inequality he u T �T � C is not fulfilled, then we proceed to Stage 3, otherwise, we proceed to Stage 4.”, why?

17- In result section, “QoS1 : C �160 , 140 u T ��, 120 he T ��, 10 u ����, 6 hu ����, 4 e ����, 3 he ����, 10 u ������, 2 e ������.”, How were these values established?

6. PLOS authors have the option to publish the peer review history of their article (what does this mean?). If published, this will include your full peer review and any attached files.

Reviewer #1: No

Reviewer #2: **Yes: **Zaid Ameen Abduljabbar

---

## [Author Response · Author response to Decision Letter 0]

14 Nov 2023

Full responses to the comments of the respected Editor and Reviewers are included in a separate document as part of the submission.

---

## [Decision Letter · Decision Letter 1]

20 Nov 2023

Evaluation of the QoS Policy Model of an Ordinary 5G Smart City Cluster with Predominant URLLC and eMBB Traffic

PONE-D-23-33802R1

Dear Dr. Kovtun,

We’re pleased to inform you that your manuscript has been judged scientifically suitable for publication and will be formally accepted for publication once it meets all outstanding technical requirements.

Kind regards,

Vincent Omollo Nyangaresi, Ph.D

Academic Editor

PLOS ONE

Additional Editor Comments (optional):

Reviewers' comments:

Reviewer's Responses to Questions

**Comments to the Author**

1. If the authors have adequately addressed your comments raised in a previous round of review and you feel that this manuscript is now acceptable for publication, you may indicate that here to bypass the “Comments to the Author” section, enter your conflict of interest statement in the “Confidential to Editor” section, and submit your "Accept" recommendation.

Reviewer #1: All comments have been addressed

Reviewer #2: All comments have been addressed

2. Is the manuscript technically sound, and do the data support the conclusions?

Reviewer #1: Yes

Reviewer #2: Yes

3. Has the statistical analysis been performed appropriately and rigorously? 

Reviewer #1: Yes

Reviewer #2: Yes

4. Have the authors made all data underlying the findings in their manuscript fully available?

Reviewer #1: Yes

Reviewer #2: Yes

5. Is the manuscript presented in an intelligible fashion and written in standard English?

Reviewer #1: Yes

Reviewer #2: Yes

6. Review Comments to the Author

Reviewer #1: (No Response)

Reviewer #2: The authors have considered all comments and made all necessary revisions well, so I recommend accepting the article

7. PLOS authors have the option to publish the peer review history of their article (what does this mean?). If published, this will include your full peer review and any attached files.

Reviewer #1: No

Reviewer #2: No

---

## [Editor Report · Acceptance letter]

22 Nov 2023

PONE-D-23-33802R1 

Evaluation of the QoS Policy Model of an Ordinary 5G Smart City Cluster with Predominant URLLC and eMBB Traffic 

Dear Dr. Kovtun:

I'm pleased to inform you that your manuscript has been deemed suitable for publication in PLOS ONE. Congratulations! Your manuscript is now with our production department. 

Kind regards, 

on behalf of

Dr. Vincent Omollo Nyangaresi 

Academic Editor

PLOS ONE